# Priming Potato Plants with Melatonin Protects Stolon Formation under Delayed Salt Stress by Maintaining the Photochemical Function of Photosystem II, Ionic Homeostasis and Activating the Antioxidant System

**DOI:** 10.3390/ijms24076134

**Published:** 2023-03-24

**Authors:** Marina V. Efimova, Elena D. Danilova, Ilya E. Zlobin, Lilia V. Kolomeichuk, Olga K. Murgan, Ekaterina V. Boyko, Vladimir V. Kuznetsov

**Affiliations:** 1Department of Plant Physiology, Biotechnology and Bioinformatics, Biological Institute, National Research Tomsk State University, Lenin Avenue 36, Tomsk 634050, Russia; 2K.A. Timiryazev Institute of Plant Physiology, Russian Academy of Sciences, Botanicheskaya Street 35, Moscow 127276, Russia

**Keywords:** *Solanum tuberosum* L., melatonin, priming, chloride salinity, ion homeostasis, proline, flavonoids, antioxidant enzymes, photosystem II, gene expression

## Abstract

Melatonin is among one of the promising agents able to protect agricultural plants from the adverse action of different stressors, including salinity. We aimed to investigate the effects of melatonin priming (0.1, 1.0 and 10 µM) on salt-stressed potato plants (125 mM NaCl), by studying the growth parameters, photochemical activity of photosystem II, water status, ion content and antioxidant system activity. Melatonin as a pleiotropic signaling molecule was found to decrease the negative effect of salt stress on stolon formation, tissue water content and ion status without a significant effect on the expression of Na^+^/H^+^-antiporter genes localized on the vacuolar (*NHX1* to *NHX3*) and plasma membrane (*SOS1*). Melatonin effectively decreases the accumulation of lipid peroxidation products in potato leaves in the whole range of concentrations studied. A melatonin-induced dose-dependent increase in Fv/Fm together with a decrease in uncontrolled non-photochemical dissipation Y(NO) also indicates decreased oxidative damage. The observed protective ability of melatonin was unlikely due to its influence on antioxidant enzymes, since neither SOD nor peroxidase were activated by melatonin. Melatonin exerted positive effects on the accumulation of water-soluble low-molecular-weight antioxidants, proline and flavonoids, which could aid in decreasing oxidative stress. The most consistent positive effect was observed on the accumulation of carotenoids, which are well-known lipophilic antioxidants playing an important role in the protection of photosynthesis from oxidative damage. Finally, it is possible that melatonin accumulated during pretreatment could exert direct antioxidative effects due to the ROS scavenging activity of melatonin molecules.

## 1. Introduction

The productivity of crops is largely determined by negative environmental factors, such as salinization, unfavorable temperatures, drought, anthropogenic soil pollution, etc. Among the damaging abiotic factors, salt stress has the greatest negative impact on plant productivity and the quality of the production obtained [1]. More than six percent of the world’s total land area and twenty-two percent of cultivated land is affected by salinity. The expansion of saline territories is currently facilitated by unfavorable global climate change and intensive urbanization [1]. Thus, in accordance with the forecast of Sharma, D.K. and Singh, A., saline areas in Southeast Asia will triple by 2050 [2].

The potential negative effects of salinity are exacerbated by the fact that the vast majority of crops, including potatoes, are glycophytes [3] and are therefore unable to grow in saline habitats. The potato (*Solanum tuberosum* L.) is the fourth-most-important food consumed in the world after rice, wheat and corn [4]. The world population increase implies a significant increase in food production, which makes it relevant to study the mechanisms of salt tolerance of *S. tuberosum* in order to find ways to increase its productivity in saline areas.

The negative effect of salt stress is based on the accumulation of inorganic ions, primarily Na^+^ and Cl^−^, in plant tissues, which results in a violation of the osmotic balance and in the toxic effect of excess salt on cellular metabolism [5,6,7,8]. Under salt stress, the plant experiences water deficiency, an imbalance of ions and intense generation of reactive oxygen species (ROS). The development of oxidative stress is accompanied by the disruption of the functioning of macromolecules and cell membranes, which leads to the inhibition of the main physiological processes, primarily photosynthesis and growth [8].

Modern agronomic and engineering technologies aimed at the recultivation of saline lands are unable to solve this problem on a global scale [9]. The use of phytohormones and other physiologically active compounds in agricultural production is a cost-effective and environmentally safe way to increase the salt tolerance of plants. In recent years, melatonin has been shown to be highly effective as a protective molecule against the damaging effects of many abiotic factors and biopathogens [10,11,12,13,14,15,16].

Melatonin (N-acetyl-5-methoxytryptamine) was discovered as a vertebrate pituitary hormone. Its presence has been established in almost all groups of living organisms, from ancient photosynthetic prokaryotes to invertebrates and vertebrates. In plants, melatonin was discovered only in 1995 [11,13]. Melatonin is a multi-functional molecule [17,18,19]. Both melatonin itself and its metabolic products (cyclic 3-hydroxymelatonin, N1-acetyl-N2-formyl-5-methoxykynuramine, N1-acetyl-5-methoxykynuramine, 6-hydroxymelatonin and 2-hydroxymelatonin) act as effective antioxidants, reducing the content of ROS or reactive nitrogen species (RNS), which leads to a decrease in the intensity of lipid peroxidation [11,20,21]. Due to its amphipathic nature, the melatonin molecule can scavenge free radicals in both hydrophilic and lipophilic environments [20,22]. Under stress conditions, melatonin regulates a number of physiological processes, including the activation of plant growth, an increase in the efficiency of photosynthesis and the accumulation of low-molecular-weight organic osmolytes [17,23,24]. It is hypothesized that melatonin acts as a plant hormone, as evidenced by the discovery in *Arabidopsis thaliana* L. plants of the CAND2/PMTR1 melatonin receptor, through which the regulation of the transpiration process is carried out due to the activation of the Gα subunit; H_2_O_2_ and Ca^2+^ are the components of the melatonin signal transduction cascade [25].

Currently, a lot of data have been accumulated on the protective role of melatonin in salt stress [15,26,27,28]. It has been shown in several species that melatonin acts under salt stress as an antioxidant and modulates plant hormone crosstalk [16,20,29]. In cucumber plants, exogenous melatonin activates antioxidant enzymes and causes the accumulation of low-molecular-weight organic compounds with osmoregulatory and antioxidant properties [30]. In addition, melatonin maintains the ionic status of plants under salt stress, especially the balance between sodium and potassium ions [15,29,30,31,32].

Almost all of the above studies were performed under conditions of simultaneous action of melatonin and stress on the plant, while plant priming by melatonin has been studied very poorly [20,22]. Priming is a mechanism which leads to a physiological state that enables plants to respond more rapidly and/or more robustly after exposure to biotic or abiotic stress in the future [33,34]. This phenomenon is realized through complex epigenetic events such as DNA methylation, ATP-dependent chromatin remodeling, histone modification and others [34,35].

The most-studied kind of priming at present is seed priming, including seed priming with melatonin, which allows the regulation of the metabolic processes of plants in the early stages of germination under stress conditions [36,37]. Priming requires a relatively short-term pretreatment of seeds, which accelerates the germination of wheat seeds under the action of chromium salts, regulates carbohydrate metabolism in cold-treated corn, increases antioxidant activity in basil and maintains the osmotic status of drought-treated rapeseed [36,38].

There are several publications on the melatonin priming of vegetative plants [39,40]. For example, long-term (for 7 days) priming of rapeseed seedlings with melatonin before the onset of drought contributed to an increase in catalase activity in plant roots, stimulated the growth of lateral roots and prevented stomata from closing in drought conditions [41]. In rice plants primed with melatonin in the form of either a leaf spray or direct root application reduced the damaging effects of low-temperature stress on photosynthesis and photosystem II activity, reduced oxidative stress and protected thylakoid membranes from damage [39]. Pretreatment of *Avena sativa* plants with melatonin protected the functioning of photosystem II from subsequent salt stress, activated the antioxidant system and stimulated the expression of genes for the main proteins of photosystem II [42]. The protective effect of long-term (from 3 to 15 days) melatonin pretreatment of cucumber, tomato and bitter melon plants under salinity stress has been demonstrated [43,44,45], and the involvement of melatonin in the regulation of K^+^/Na^+^ homeostasis and lipid metabolism in salinized sweet potato has been shown [46]. However, the question of the identification of the physiological mechanisms of the possible protective effect of short-term melatonin priming of potato plants against subsequent salt stress remained open.

In this regard, the aim of our study was to find out whether the short-term priming of potato plants by melatonin exerted protective effects on growth processes and stolon formation during subsequent salinity stress. We also aimed to study whether melatonin was involved in the regulation of the photochemical function of photosystem II, the cellular antioxidant system and ionic balance.

## 2. Results

### 2.1. The Growth Parameters of Potato Plants

The following growth parameters of potato plants were assessed: the linear dimensions of the axial organs, the total area of the assimilating surface, the number of leaves and stolons on the seventh day after the start of exposure (1 day of melatonin, 6 days of 125 mM NaCl) (Table 1). Chloride salinity inhibited the growth of axial organs, reduced the total leaf surface area and, ultimately, the total weight of plants by 20, 66 and 40%, respectively (Table 1). The most prominent negative effect of salinity was observed for stolon formation, since the number of stolons decreased by 6 times under conditions of salt stress. Short-term treatment of potato plants with melatonin, regardless of the effective concentration, significantly reduced the negative effect of 125 mM NaCl on stolon formation, but did not affect other morphometric parameters studied (Table 1).

### 2.2. The Content of Photosynthetic Pigments in Potato Leaves

The content of chlorophylls *a*, *b* and carotenoids decreased by 2.0–2.4 times in response to salt stress (Figure 1). Due to the uniform decrease in all studied groups of photosynthetic pigments under salinity, the Chl *a*/Chl *b* and Chl/Car ratios changed insignificantly. Short-term treatment of potato plants with melatonin removed the negative effect of the stressor partially or completely; the minimum protective effect was shown for a hormone concentration of 0.1 μM, and the most prominent protective effect was observed at 1 μM melatonin (Figure 1).

### 2.3. Primary Photosynthetic Processes in Potato Leaves

The indicators of photochemical activity of photosystem II of potato plants under salinity are shown in Figure 2 and in Appendix A. In control plants, the maximum quantum yield of photosystem II (Fv/Fm) was 0.81 (Figure 2), which corresponds to the Fv/Fm values characteristic of plants not subjected to stress [47]. Salt treatment induced a strong decrease in Fv/Fm down to 0.47, but 1 μM and 10 μM melatonin pretreatment partially alleviated the negative NaCl effect (Figure 2). Similar changes were observed for the effective quantum yield (Y(II)) and electron transport rate (ETR), which decreased by approximately 40% under saline conditions relative to the control, while under treatment with 1 μM melatonin, the Y(II) values and ETR differed from the values of the control variant by only 15% (Figure 2).

In parallel with a decrease in the ETR, Fv/Fm and Y(II), the quantum yields of controlled (Y(NPQ)) and uncontrolled (Y(NO)) non-photochemical dissipation of light energy increased in salt-stressed plants by 86% and 45%, respectively. Pretreatment of plants with 10 µM melatonin contributed to the maintenance of the Y(NO) at the control level, but at the same time had no significant effect on the Y(NPQ) value.

### 2.4. Water Content and Leaf Cell Sap Osmotic Potential

The water content in the leaves and roots of the control plants was 89.85% and 94.60%, respectively. In response NaCl treatment, it decreased by 6.02% and 2.61%, respectively (Figure 3). Melatonin at all concentrations used partially eliminated the negative effect of NaCl in leaf tissues, but not in root tissues (Figure 3).

The osmotic potential of the content of potato leaf cells under control conditions was 0.79 MPa (Figure 4). Plants responded to the action of NaCl by a twofold decrease in the osmotic potential of the content of leaf cells. Pretreatment of plants with melatonin did not have a significant effect on the value of the osmotic potential of leaves and, therefore, did not increase the water absorption function of plants under salt stress.

### 2.5. Lipid Peroxidation u Antioxidant Enzyme Activity in Potato Leaves

Under control conditions, the content of TBARS in the leaves was 0.102 μM/g fresh weight and salinity increased TBARS content by 16%. Melatonin reduced the content of the formed TBARS to the control level, regardless of the active concentration (Figure 5).

A slight increase in the level of lipid peroxidation during salinity was accompanied by a slight increase (by 27%) in the SOD activity (Figure 6A), while the activity of guaiacol-dependent peroxidase did not change significantly (Figure 6B). Plant pretreatment with melatonin did not affect NaCl-dependent SOD activation. At the same time, 10 µM melatonin decreased peroxidase activity by a factor of two relative to the control (Figure 6B). Notably, melatonin pretreatment of potato plants with subsequent salinization inhibited the expression of the *APX1* ascorbate peroxidase gene, but did not affect the expression level of its homolog *APX2* (Appendix A).

### 2.6. Proline and Flavonoid Content in Potato Leaves

Under control conditions, the content of proline in the leaves was about 4.84 µM/g fresh weight. In response to the action of 125 mM NaCl, plants demonstrated a significant (8.1-fold) accumulation of proline. Pretreatment of potato plants with 10 µM melatonin increased the NaCl-induced proline level up to 11 times higher than in the control (Figure 7).

The accumulation of proline under saline conditions was accompanied by a significant (by 30%) increase in the levels of transcripts of both the proline precursor synthesis gene, pyrroline-5-carboxylate synthetase (*P5CS1*) and the proline degradation gene, proline dehydrogenase (*PDH*) (Figure 8). Melatonin priming of potato plants had little effect on *P5CS1* gene expression, but significantly reduced *PDH* expression, which could be responsible for the observed accumulation of proline. The content of flavonoids was not influenced by salinity, but was increased by priming with 0.1 µM and 1 µM melatonin (Figure 9).

Evaluation of transcript levels of the key enzymes of phenylpropanoid metabolism phenylalanine lyase (*PAL*) and chalcone synthase (*CHS1α*) showed that potato plants responded to salt stress with some increase in PAL gene expression intensity, while priming plants with melatonin followed by salinization inhibited the stimulating effect of salt. In contrast, *CHS1α* gene expression was dramatically suppressed under salt stress. Pretreatment of plants with melatonin did not prevent the inhibitory effect of salt on the transcript levels of this gene (Appendix A). The increase in flavonoid levels in potato plants under salt stress with the inhibition of CHS1α expression can be explained by melatonin inhibition of flavonoid degradation under these conditions.

### 2.7. Effect of Salt Stress and Melatonin on the Content of Sodium and Potassium Ions in Roots, Stems and Leaves of Potato Plants

Inorganic ions are major contributors to the osmotic potential of the cell exudate, but when they reach a certain concentration, they cause osmotic stress and have a toxic effect on plant metabolism. We analyzed the content of sodium, chlorine and potassium ions in the leaves, stem and roots of plants. As follows from the data obtained, in the absence of a stressor the minimum content of sodium ions was noted in leaves, the maximum content of sodium ions was noted in the stem; the content of potassium ions and chlorine did not show obvious organ specificity (Table 2). Under chloride salinity, plants actively accumulated inorganic ions. The content of Na^+^ increased in the leaves, stem and root by 74, 32 and 26 times, respectively, while the content of Cl^−^ increased in the same organs by 7, 14 and 5 times, respectively. Potassium ion content decreased in the leaves, stem and root by 1.7, 1.4 and 1.3 times, respectively (Table 2).

Pretreatment of potato plants with melatonin resulted in reduced accumulation of sodium ions in all organs; the maximum effect of melatonin was observed at concentrations of 1.0 and 10.0 µM (Table 2). The effect of the pretreatment of plants with melatonin on chloride ion content was poorly pronounced, although there was a tendency for chloride ions to decrease, primarily in the stem and, to a lesser extent, in the root. Melatonin at a concentration of 10.0 µM slightly reduced the observed drop in potassium ions in the above-ground organs under salt stress (Table 2).

An evaluation of the ability of melatonin to affect the selective inter-organ transfer of sodium and potassium ions (S_K^+^ Na^+^_ = K^+^/Na^+^ in the leaves (or stems)/(K^+^/Na^+^ in the roots) showed that melatonin priming followed by salinization increased the K^+^ content in the leaves and stem or reduced the Na^+^ content in the root system (Table 3).

### 2.8. Effect of Melatonin on Na^+^/H^+^ Antiporter Gene Transcript Levels of the Tonoplast (NHX1to NHX3) and Plasmalemma (SOS1) in Solanum Tuberosum Plants under Saline Conditions

The regulation of intracellular sodium and potassium ion homeostasis can be achieved through the functioning of Na^+^/H^+^ antiporters localized on the vascular (*NHX1* to *NHX3*) and plasma (SOS1) membranes. The results showed that of the three *NHX* genes, the first member of this family (*NHX1*) was the most actively expressed under optimal growth conditions. In response to salt stress, plants reacted by a 2.0 to 2.5-fold increase in *NHX2* and *NHX3* gene transcript levels, while the expression of *NHX1* gene did not change. Priming of plants with melatonin followed by salinization decreased the intensity of *NHX1* gene expression but did not affect the levels of *NHX2* and *NHX3* gene transcripts initiated by salt stress. The *SOS1* gene, encoding the Na^+^/H^+^ antiporter of plasma membrane, as well as the *NHX2* and *NHX2* genes were activated in response to 125 mM NaCl. At the same time, the pretreatment of plants with melatonin at a concentration of 10 µM resulted in an additional increase in *SOS1* gene transcript levels (Table 4).

## 3. Discussion

### 3.1. Potato Stolon Formation and Plant Water Status

When the salt content in the soil (nutrient) solution increases, there is a sharp drop in its water potential, which leads to a slowdown in the water absorption capacity of the root system and the development of water deficiency in plants [48]. Under conditions of osmotic stress, the intensity of transpiration is reduced and growth processes, especially those of above-ground organs, are inhibited. Potato plants reacted to the salt stress (125 mM NaCl, 6 days) by a two-times lower osmotic potential of leaf cells content (Figure 4), by significant tissue dehydration (Figure 3) and by growth retardation of axial organs, leaf surface area and total weight (Table 1). Interestingly, the process that was the most sensitive to salt stress was the process of stolon formation, the number of which decreased by 80% of the control (Table 1). This type of plant response to salt stress is typical for the potato plant [49]. Melatonin pretreatment (0, 01, 1.0, 10 mlM) had no protective effect on the assessed growth parameters, had no effect on the osmotic potential value (Figure 4), but slightly increased the leaf water content (Figure 3) and significantly reduced the negative effect of salt on stolon formation (Table 1). Considering that high sucrose concentration is required for stolon formation and microtubers, it can be assumed that melatonin priming of plants restores sugar translocation impaired by salt stress [50]. The inability of melatonin priming to reduce the subsequent negative effects of salt stress on potato plant growth is obviously species-specific, and may depend on the physiological status of the plant, the intensity and duration of the salt stress and the method of hormonal treatment, as the pretreatment of oat plants prior to salt stress [51], the tomato plant prior to exposure to low temperature and salinity [52,53] and *Pisum sativum* L. seedlings prior to paraquat (PQ)-induced oxidative stress [54] was accompanied by a reduction in the damaging effect of stressors on growth processes. The experimental data we obtained do not allow us to exclude that exogenous melatonin will show a salt-protective effect on potato plants at a different stage of ontogenesis or under different experimental conditions.

### 3.2. Contents of Sodium, Chlorine and Potassium Ions in Leaves, Stem and Roots of Potato Plants

Salt stress increased the content of sodium ions in potato leaves, stem and root by 74, 32 and 26 times, respectively, while the content of chlorine ions increased in the same organs by 7, 14 and 5 times (Table 2). The accumulation of sodium ions to a certain level is accompanied by their toxic effect on cellular metabolism, which is manifested by the inhibition of enzyme activity, disruption of cellular macromolecular structures, membrane integrity, ion homeostasis and the generation of reactive oxygen species [49]. The reactive oxygen species disturbed the vital functions of potato plants [55]. Along with Na^+^ and Cl^−^ accumulation, a 1.7, 1.4 and 1.3-fold decrease in K^+^ in the leaves, stem and root, respectively, was observed (Table 2). K^+^ is essential for plant growth and development [56]. In addition to inhibition of K^+^ uptake under salt stress, the uptake of magnesium ions, which is a part of the chlorophyll molecule and is a cofactor of a number of chlorophyll biosynthesis enzymes, was inhibited [57].

Maintenance of ion homeostasis under conditions of salt stress is achieved by the transport and accumulation of sodium ions in metabolically less-active organs and tissues, for example, in old leaves and the stem [49]. Pretreatment of potato plants with melatonin (1.0 and 10.0 µM) under subsequent salinization resulted in a reduction of sodium ion accumulation in all organs (Table 2). The effect of the pretreatment of plants with melatonin on chlorine ion content was poorly pronounced, although there is a tendency for a reduction in chlorine ions, primarily in the stem and, to a lesser extent, in the root. Melatonin at a concentration of 10.0 µM slightly reduced the observed drop in potassium ions in the above-ground organs under salt stress (Table 2). Similar results were obtained on bitter lemon plants in which priming with melatonin (150 µM) led to a decrease in Na^+^ content but an increase in K^+^, Ca^2+^ and P content under delayed salt stress [44]. An estimate of the ratio of K^+^/Na^+^ in the leaves (stem) to the ratio of these two ions in the root shows that melatonin priming of plants was accompanied by an increase in K^+^ in leaves and stem compared to the root or a decrease in Na^+^ in the potato root system (Table 3).

The regulation of intracellular sodium and potassium ion content can also be achieved by the functioning of Na^+^/H^+^ antiporters localized on vacuolar (NHX1 to NHX3) and plasma (SOS1) membranes [32]. This allows sodium ions to be sequestered and stored in the vacuole or exported from the cytoplasm to the apoplast, which lowers the level of sodium ions in the cytoplasm and reduces its toxic effect on metabolism [49]. The results showed that plants responded to salt stress by increasing the levels of *NHX2* and *NHX3* gene transcripts, while *NHX1* gene expression did not change. Priming plants with melatonin reduced the intensity of *NHX1* gene expression but did not affect the NaCl-dependent expression of *NHX2* and *NHX3* genes. The *SOS1* gene encodes the Na^+^/H^+^ antiporter of plasma membrane that was activated under salt stress. Pretreatment of plants with melatonin (10 µM) increased its transcript levels, but this effect of melatonin was not statistically significant (Table 4). These data indicate that in potato plants melatonin probably stimulates the functioning of the SOS1 antiporter by exporting sodium ions from the cytoplasm to the apoplast. In relation to the issue under discussion, interesting data were published by [46], according to which exogenous MT (melatonin) stimulated TAG (triacylglycerol) breakdown, FA (fatty acid) β-oxidation and energy turnover under salinity conditions, thereby contributing to the maintenance of PM (plasma membrane) H+-ATPase activity and K^+^/Na^+^ homeostasis in sweet potato.

### 3.3. Basic Photosynthetic Pigments and Photochemical Activity of Photosystem II

Plant growth and the formation of stolons and microtubers in potatoes are directly dependent on the ability of the photosynthetic apparatus to assimilate carbon dioxide and synthesize carbohydrates. There is no doubt that salt stress reduces the intensity of photosynthesis by damaging the photosynthetic pigment system [58]. It is known, for example, that the stomata closure increases under salinity stress, thereby reducing the CO_2_ uptake through the stomatal pore and thus causing a reduction in photosynthesis. The rate of photosynthesis per unit area was reduced due to a reduction in stomata number in potatoes [59]. Stress conditions destroy the chloroplast ultrastructure and lead to a decrease in chlorophyll, which results in lower photosynthetic activity [60].

Our data show that the content of the main photosynthetic pigments, chlorophylls *a*, *b* and carotenoids, decreased by a factor of 2.0 to 2.4 in response to salt stress (Figure 1). Priming potato plants with melatonin partially or completely abolished the negative effect of delayed salt stress on these pigments (Figure 1). This effect of melatonin could be due to an acceleration of pigment synthesis or an inhibition of the activity of chlorophyllase, pheophytinase and peroxidase enzymes responsible for chlorophyll degradation [61]. In addition, melatonin could reduce the negative effect of salinity on chlorophyll content by activating the cellular antioxidant system (Figure 5 and Figure 9) and reducing ROS levels that disrupt the synthesis of these pigments. It is not only the pigments but also two photosystems (PS I and PS II) that play a vital role in maintaining photosynthesis. PS II is a sensitive target of salt stress and is strongly suppressed under conditions of excessive salinity [40]. As follows from the data presented (Figure 2, Appendix A), the values of maximal quantum yield of PS II (Fv/Fm), effective quantum yield (Y(II) and electron transport rate (ETR) decreased under salt stress conditions, while melatonin significantly reduced the negative effect of salt (Figure 2). Under saline conditions the quantum yields of regulated (Y(NPQ)) and unregulated (Y(NO)) non-photochemical light energy dissipation increased by 86% and 45%, respectively, relative to the control values. Priming plants with melatonin (10 μM) under delayed salt stress helped to maintain Y(NO) at control values but had no effect on Y(NPQ). In tomato plants, melatonin priming was also shown to reduce the damage to the photosynthetic apparatus and increase the electron transfer rate and quantum yield of PSI and PSII photochemistry, to protect the thylakoid membrane from damage caused by low-temperature stress [53]. A positive effect of melatonin priming on photosynthesis under subsequent salinization was demonstrated in oat plants. The kinetic analysis of chlorophyll fluorescence showed that almost all chlorophyll fluorescence parameters improved under the influence of melatonin. Moreover, the expression of genes encoding the major PSII external proteins (PsbA, PsbB, PsbC and PsbD) was strongly activated after melatonin priming [42]. Similar results on the protective role of melatonin photochemical PSII activity under salt stress were obtained in bitter melon plants [43] and tomato plants [45]. Stress tolerance of photosynthetic processes can be initiated not only by melatonin but also by the pretreatment of plants with factors of different physical and chemical natures [61].

### 3.4. Components of Antioxidant System and Proline Accumulation

One of the universal plant responses to stress is the generation of ROS, which damage membranes, macromolecules, cellular metabolism and physiological functions. Melatonin is an effective antioxidant. It regulates the activity of enzymatic and non-enzymatic antioxidants that can reduce ROS levels under stress and can reduce oxidative stress by itself, either by directly inactivating free radicals or by slowing the generation of reactive oxygen species [62,63].

We observed that melatonin effectively decreases the accumulation of lipid peroxidation products in potato leaves in the whole range of concentrations studied (Figure 5). A melatonin-induced dose-dependent increase in Fv/Fm together with a decrease in uncontrolled non-photochemical dissipation Y(NO) (Figure 2) also indicates the decreased oxidative damage, since PSII is highly susceptible to inhibition by ROS [64]. The observed protective ability of melatonin was unlikely due to its influence on the antioxidant enzymes, since neither SOD nor peroxidase were activated by melatonin. Melatonin exerted positive effects on the accumulation of water-soluble low-molecular-weight antioxidants, proline and flavonoids, which could aid in decreasing oxidative stress [65]. However, these positive effects were not universal across the range of melatonin concentrations used (Figure 7 and Figure 9). The most consistent positive effect was observed in the accumulation of carotenoids, which are well-known lipophilic antioxidants playing an important role in the protection of photosynthesis from oxidative damage. Finally, it is possible that melatonin accumulated during the pretreatment could exert a direct antioxidative effect due to the ROS scavenging activity of melatonin molecules. However, we were unable to test this hypothesis.

## 4. Materials and Methods

### 4.1. Plant Growth and Experimental Design

The experiments were conducted on potato (*Solanum tuberosum* L.) plants, cv. Lugovskoi (identifier 8301891). The plants were regenerated in vitro from an apical meristem and cultivated on half-strength MS agar medium for 30 days. Then, 30-day seedlings were adapted to liquid half-strength MS medium for two weeks in a controlled-climate chamber under L36 W/77 Fluora luminescent lamps (Osram, Munich, Germany) at a PAR quantum flux density of 200–250 μmol photons × m^−2^ × s^−1^) with a 16 h photoperiod and day/night temperatures of 23 ± 0.5 °C/20 ± 0.5 °C. After 14 days of adaptation to liquid medium, the plants were transferred either to the medium with melatonin (0.1, 1, 10 μM) or to the control medium for 24 h. Next, untreated plants were transferred to either the control medium or a medium with 125 mM NaCl (“control” and “NaCl” variants) and treated plants were transferred to the medium with 125 mM NaCl but without melatonin. The roots of these plants were washed thoroughly with water beforehand to remove absorbed melatonin from the root surface. Photosynthetic and biochemical parameters were determined in the leaves of potato seedlings 6 days after the beginning of the NaCl treatment. The effective concentration of NaCl was determined earlier [66]; concentrations of melatonin were taken from a previous study (unpublished date).

### 4.2. Determination of Growth Parameters

The linear dimensions of the axial organs, the total area of the leaf surface, the number of nodes and stolons of potato plants were determined.

### 4.3. Determining the Fresh and Dry Weight, and Water Content

Fresh and dry plant biomasses were determined gravimetrically. The fresh weights of the plants were determined with an accuracy of 1 mg using an analytical balance (Scout Pro SPU123, Ohaus Corporation, Parsippany, NJ, USA). The dry weight was determined using an analytical balance (AB54-S, Mettler Toledo, Switzerland) with an accuracy of 0.1 mg after drying the samples to a constant weight at 70 °C. The water content is expressed as a percentage of fresh weight.

### 4.4. Determining the Osmotic Potential in Potato Plants

The osmotic potential of cell exudates was determined using an Osmomat 030 cryoscopic osmometer (Gonotec, Berlin, Germany) according to the manufacturer’s instructions. Cell sap was squeezed from defrosted leaf samples.

### 4.5. Determination of Photosynthetic Pigments

The chlorophylls *a* (Chl *a*), *b* (Chl *b*) and carotenoids (Car) contents were determined using the Lichtenthaler method [67]. To determine the content of photosynthetic pigments, a sample of leaves (15 mg) was ground in 96% ethanol and the homogenate was centrifuged for 10 min at 8000 rpm using a MiniSpin centrifuge (Eppendorf, Hamburg, Germany). The optical density of the alcoholic extract (the final volume of the solution was 1.5 mL) was measured using a Genesys 10S UV-Vis spectrophotometer at wavelengths of 470, 648, 664 and 720 nm. The pigment concentration in the alcoholic extract was calculated according to Lichtenthaler formulas [67].

### 4.6. Determination of Chlorophyll Fluorescence

The photochemical activity of photosystem II (PSII) was measured with a PAM fluorimeter (Junior-PAM, Heinz-Walz, Germany). Before measurement, the samples were adapted to darkness for 20 min. Then, the light was switched on for 10 min (I = 190 μmol (quantum) m^−2^ s^−1^ PAR). The intensity of the saturating light was 6000 μmol (quantum) m^−2^ s^−1^. Saturating pulses were generated every 60 s. The parameter calculations on the basis of fluorescence data were performed using WinControl-3 v.3.32 software (Walz, Effeltrich, Germany). The values for F_0_, F_v_, F_m_, F_m_’ and F_0′_, as well as the PSII maximum (F_v_/F_m_) and effective Y(II) (F_m_’-F_t_)/F_m_’ photochemical quantum yields and non-photochemical quenching (NPQ) (F_m_/F_m_’-1), were determined. F_m_ and F_m_’ are the maximum Chl fluorescence levels under dark- and light-adapted conditions, respectively. F_v_ is the photoinduced change in fluorescence and F_t_ is the level of fluorescence before a saturation impulse is applied. F_0_ is the initial Chl fluorescence level. The quenching parameters were also determined: NPQ—non-photochemical fluorescence quenching; Y(NO)—quantum yield of non-regulated non-photochemical energy dissipation in PSII and Y(NPQ)—quantum yield of regulated non-photochemical energy dissipation in PSII, Y(NO) + Y(NPQ) + Y(II) = 1.

### 4.7. Evaluating of the Lipid Peroxidation Level

The level of lipid peroxidation was evaluated by the formation of a colored complex between thiobarbituric acid and thiobarbituric-acid-reactive substances (TBARS) upon heating [68]. TBARS content was determined spectrophotometrically with a Genesys 10S UV-Vis Genes spectrophotometer (Thermo Fisher Scientific, Waltham, MA, USA) at wavelengths of 532 and 600 nm.

### 4.8. Determination of Proline Content

Free proline was extracted and determined as previously described [69]. The proline content was measured with a Genesys 10S UV-Vis Genes spectrophotometer (Thermo Fisher Scientific, USA) at a wavelength of 520 nm.

### 4.9. Determination of the Total Content of Flavonoids

The determination of the total content of flavonoids was carried out according to Gage with modifications [70] A sample of plants (200–300 mg) was crushed and transferred to a 250 mL flask, 100 mL of 70% ethanol was added, a reflux condenser was attached and heated in a boiling water bath for 60 min. After cooling, the solution was filtered into a volumetric flask (flask volume 200 mL). The extraction was repeated twice under the same conditions with 50 mL of 70% ethanol for 60 min in the second phase contact and 30 min in the third phase. The volume of the combined solution was brought up to the mark with 70% ethanol (solution A). A total of 2 mL of solution A was transferred into a volumetric flask with a capacity of 25 mL, 2 mL of a 2% solution of aluminum chloride in 95% ethyl alcohol and 1 drop of 5% acetic acid were added and the volume of the solution was adjusted to the mark with 95% ethyl alcohol (solution B). The optical density of solution B was determined after 40 min on a spectrophotometer at a wavelength of 414 nm. The total content of flavonoids in terms of rutin and absolutely dry raw materials as a percentage (*X*) was calculated by the formula:$$X=\frac{D*K^{V}}{m}*\frac{m_{S}}{D^{S}*K_{S}^{V}}*\frac{100}{100-W}*100$$
*D*—the optical density of the test solution;*D^S^*—the optical density of the rutin solution;*m*—the mass of raw materials, g;*m_S_*—the mass of rutin, g;$K^{V}$—the dilution factor of the test solution (1250);$K_{S}^{V}$—dilution factor of rutin solution (2500);*W*—weight loss during drying of raw materials, %.

### 4.10. Determination of the Activity of Antioxidant Enzymes

Total superoxide dismutase (EC 1.15.1.1) and peroxidase (EC 1.11.1.7) activities were determined in crude extracts of leaf tissues. Leaf samples (200 mg) were ground in liquid nitrogen with insoluble polyvinyl pyrrolidone, extracted in 0.066 M potassium-phosphate buffer (pH 7.4) containing 0.5 mM dithiothreitol and 0.1 mM phenylmethylsulfonyl fluoride in dimethyl sulfoxide, and then centrifuged for 20 min at 8000 rpm and 4 °C using a 5430R centrifuge (Eppendorf, Germany). The total SOD activity was determined according to Beauchamp and Fridovich [71]. The reaction medium (2 mL) contained 10 μL of supernatant, 1.75 mL of 50 mM Tris-HCl buffer (pH 7.8), 0.2 mL of 0.1 M DL-methionine, 0.063 mL of 1.7 mM Nitro Blue tetrazolium (Fermentas, Waltham, MA, USA), 0.047 mL of 1% Triton X-100 and 0.060 mL of 0.004% riboflavin. The reaction proceeded under LED lamps (I = 232 μmol photons/m^−2^s^−1^) for 30 min. The absorption was measured at 560 nm using a Genesys 10S UV-Vis spectrophotometer (Thermo Fisher Scientific, USA). Peroxidase activity was determined as previously described [72]. The reaction mixture contained 50 μL of supernatant, 1.95 mL of 0.066 M potassium-phosphate buffer (pH 7.4), 200 μL of 7 mM guaiacol and 250 μL of 0.01 M H_2_O_2_. The absorption was measured at 470 nm using a Genesys 10S UV-Vis spectrophotometer (Thermo Fisher Scientific, Waltham, MA, USA).

### 4.11. The Determination of Total Protein Content

The protein content in the samples was determined according to Esen [73].

### 4.12. Analysis of Elemental Composition of Potato Plants

The leaves, shoot and root samples were digested in glass tubes with solutions of concentrated HNO_3_ and HClO_4_ (2:1 *v/v*) for 24 h at room temperature and then incubated in a dry block thermostat at 150 °C for 1.5 h and then at 180 °C for 2 h. To desorb metal ions from the apoplast, the roots were rinsed in 2 mM CaCl_2_ before digestion. An analysis of the contents of Na^+^, K^+^, Cl^−^ in the roots, shoots and leaves of potato plants was performed by inductively coupled plasma mass spectrometry (ICP-MS), Agilent 7900, Santa Clara, CA, USA.

The ability of plants to selectively translocate ions (K^+^, Na^+^) was calculated according to the formula [64]:S_K^+^,Na^+^_ = (K^+^/Na^+^ in the shoots or leaves)/(K^+^/Na^+^ in the roots)

### 4.13. RNA Isolation and cDNA Synthesis

Leaf tissues were grinded in liquid nitrogen and total RNA extraction was performed using TRIzol reagent (Invitrogen, Waltham, MA, USA) according to the manufacturer’s instructions. Genomic DNA contamination was removed by treatment with DNase I, RNase-free (Thermo Fisher Scientific, USA). cDNA synthesis was performed using the MMLV-RevertAid transcriptase (Thermo Fisher Scientific, USA) and the oligo (dT) 21 primer (Evrogen, Moscow, Russia).

### 4.14. The Selection of Target Genes for qRT-PCR Analysis and Primer Design

Primer sequences for references and PCR reaction conditions were taken from Nicot et al. [74]. Nucleotide sequences for proline metabolism genes (P5CS1 and PDH), flavonoid synthesis gene (PAL and CHS1a), antioxidant enzyme genes (APX1 and APX3) and ion transporter genes (SOS1, NHX1, NHX2 and NHX3) were taken from the NCBI database (http://www.ncbi.nlm.nih.gov/ 20 February 2023). Gene-specific primers were constructed using Primer-BLAST (https://www.ncbi.nlm.nih.gov/tools/primer-blast/ 20 February 2023) and Vector NTI 11.0 program. Primer sequences, accession numbers of target genes and sizes of PCR amplification products are given in the Appendix A.

### 4.15. Quantitative RT-PCR Analysis of Target Gene Expression

qRT-PCR analysis was performed with a thermocycler Lightcycler‘96 (Roche, Basel, Switzerland). The reaction mixture was prepared using qPCRmix HS SYBR (Eurogen, Russia), gene-specific primers, ddH_2_O and cDNA samples, according to the manufacturer’s instructions. Each reaction was performed in triple analytical replicates. For the normalization of the expression levels of the target genes, two reference genes were used, namely *cyclophilin* gene and *elongation factor 1-α* gene (*Ef1α*), as described in Nicot et al. [74]. Standard curves were obtained using serial 10-fold dilutions of cDNA samples.

### 4.16. Statistical Analysis

Each experiment was repeated at least three times. The number of plants per biological replicate ranged from nine to twelve. The figures and tables present the mean values and their standard errors (SE). The means were compared with control values at corresponding time points using the Student’s t-test. Asterisks or # represent significance at *p* ≤ 0.05.

## 5. Conclusions

When applied to salt-stressed potato plants, exogenous melatonin exerted clear stress-protective effects at the molecular level, maintaining plant water status and photosynthetic activity and increasing the accumulation of stress-protective compounds. However, these protective effects did not result in the maintenance of growth parameters of melatonin-treated plants compared to non-treated ones. The only clear exception was the formation of stolons, which was enhanced greatly by melatonin treatment. We can speculate that the protective effects of melatonin at the molecular level aided the maintenance of the carbohydrate supply, which maintained stolon formation and hence potential tuber productivity. More in-depth studies of potato-carbon balance under salt stress and melatonin treatment are required to test this hypothesis.

## Figures and Tables

**Figure 1 ijms-24-06134-f001:**
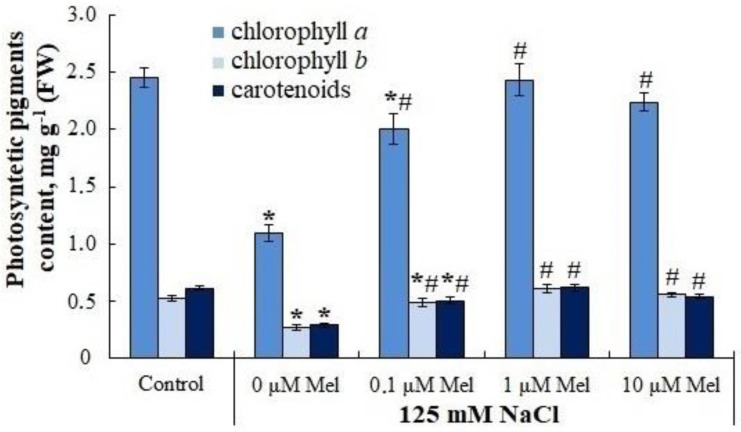
Influence of melatonin and chloride salinity on photosynthetic pigments content in *Solanum tuberosum* leaves. Asterisks (*) indicate significant differences from the control variant and the «#» indicates significant differences from the «125 mM NaCl» variant (*p* < 0.05).

**Figure 2 ijms-24-06134-f002:**
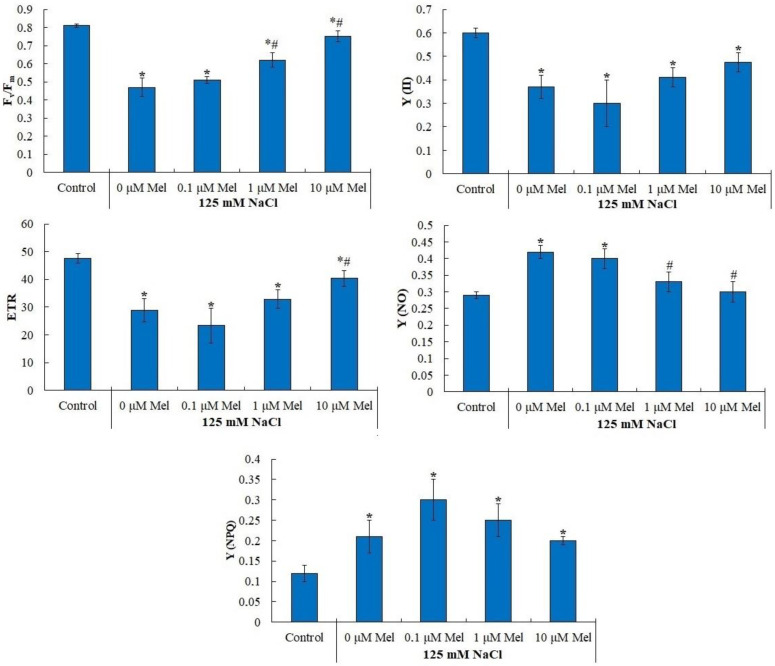
Influence of melatonin and chloride salinity on the photochemical activity of photosystem II of *Solanum tuberosum* leaves. Asterisks (*) indicate significant differences from the control variant and the «#» indicates significant differences from the «125 mM NaCl» variant (*p* < 0.05).

**Figure 3 ijms-24-06134-f003:**
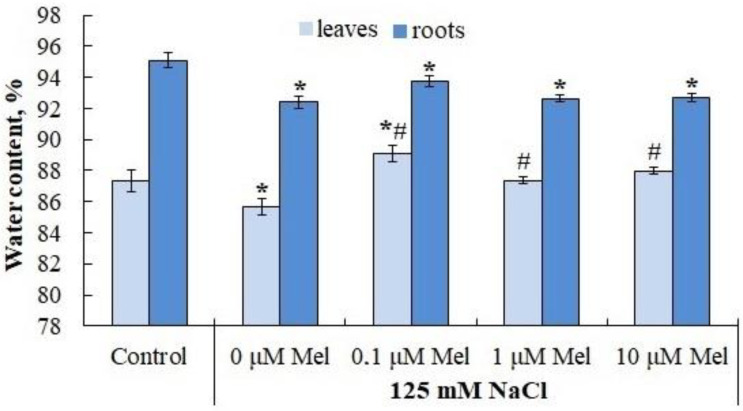
Influence of melatonin and chloride salinity on the water content (% of fresh weight) in leaves and roots of *Solanum tuberosum* plant. Asterisks (*) indicate significant differences from the control variant and the «#» indicates significant differences from the «125 mM NaCl» variant (*p* < 0.05).

**Figure 4 ijms-24-06134-f004:**
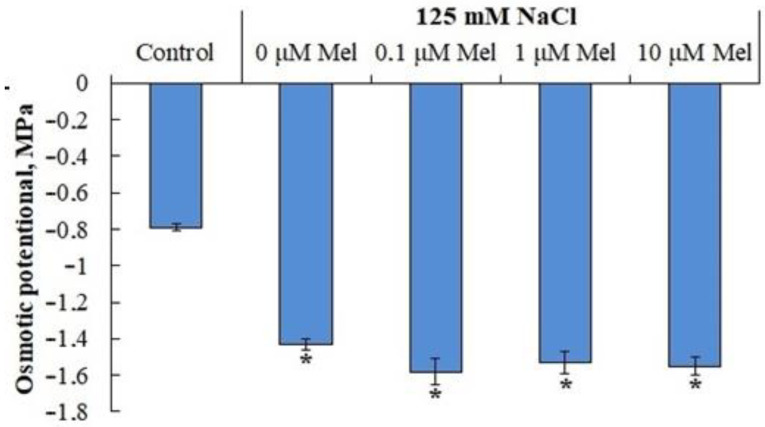
Influence of melatonin and chloride salinity on the value of the osmotic potential of the cellular exudate of *Solanum tuberosum* leaves. Asterisks (*) indicate significant differences from the control variant (*p* < 0.05).

**Figure 5 ijms-24-06134-f005:**
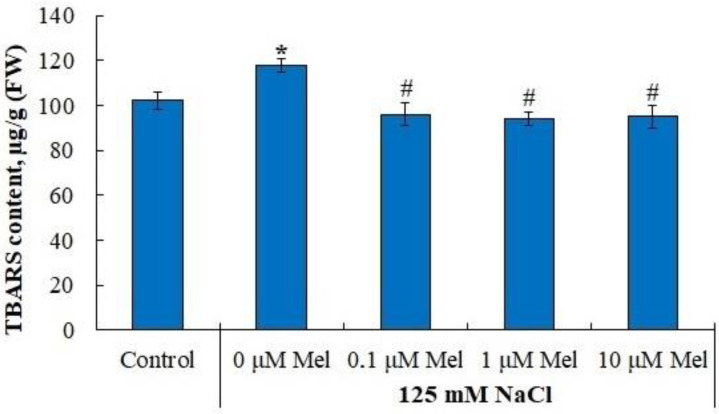
Influence of melatonin and chloride salinity on TBARS content in *Solanum tuberosum* leaves. Asterisks (*) indicate significant differences from the control variant and the «#» indicates significant differences from the «125 mM NaCl» variant (*p* < 0.05).

**Figure 6 ijms-24-06134-f006:**
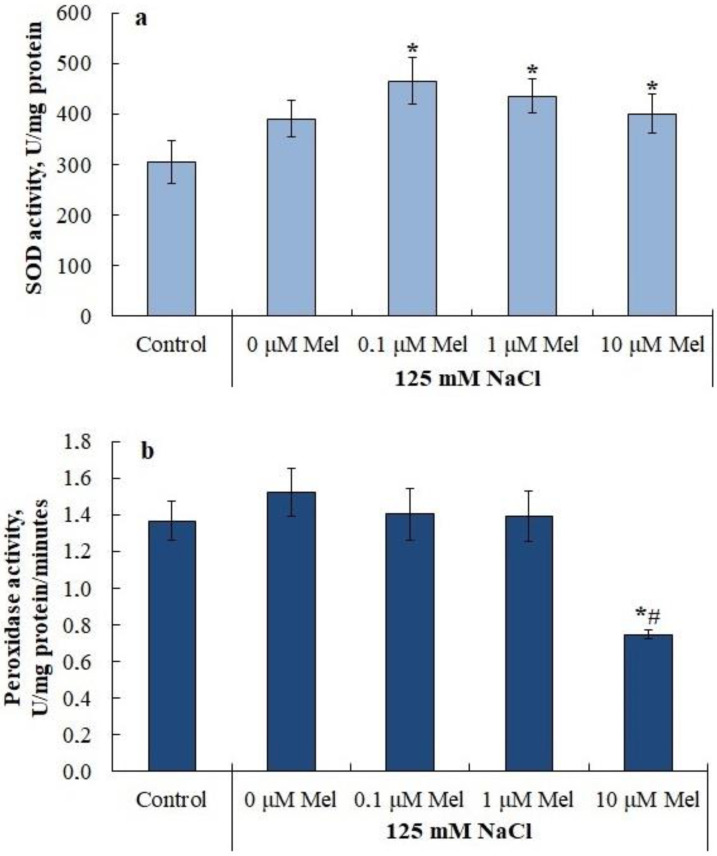
Influence of melatonin and chloride salinity on the activity of antioxidant enzymes: (**a**) Superoxide dismutase (SOD) activity; (**b**) Peroxidase activity. Asterisks (*) indicate significant differences from the control variant and the «#» indicates significant differences from the «125 mM NaCl» variant (*p* < 0.05).

**Figure 7 ijms-24-06134-f007:**
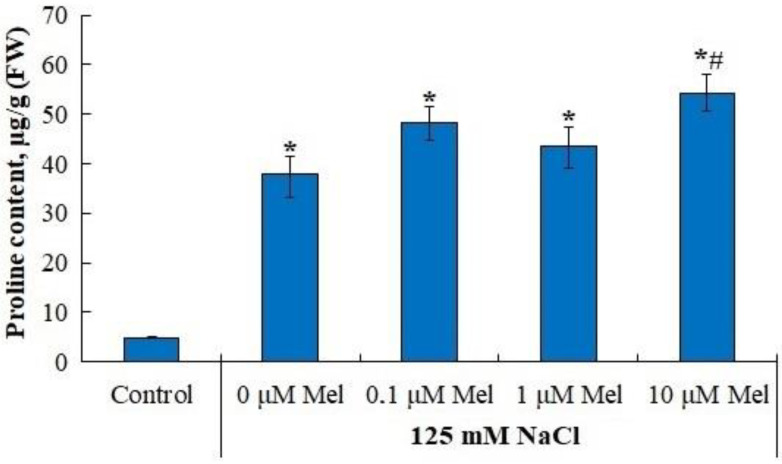
Influence of melatonin and chloride salinity on proline content in *Solanum tuberosum* leaves. Asterisks (*) indicate significant differences from the control variant and the «#» indicates significant differences from the «125 mM NaCl» variant (*p* < 0.05).

**Figure 8 ijms-24-06134-f008:**
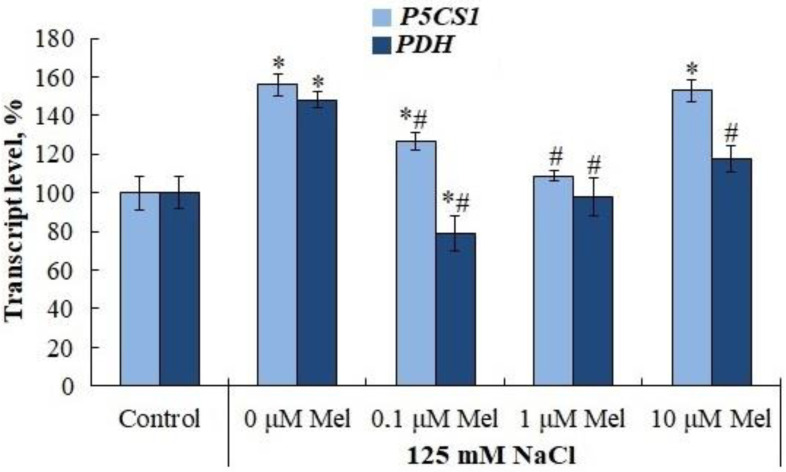
Influence of melatonin and chloride salinity on the level of transcripts of the key genes of biosynthesis (*P5CS1*) and degradation (*PDH*) of proline in *Solanum tuberosum* leaves (% of the control). Asterisks (*) indicate significant differences from the control variant and the «#» indicates significant differences from the «125 mM NaCl» variant (*p* < 0.05).

**Figure 9 ijms-24-06134-f009:**
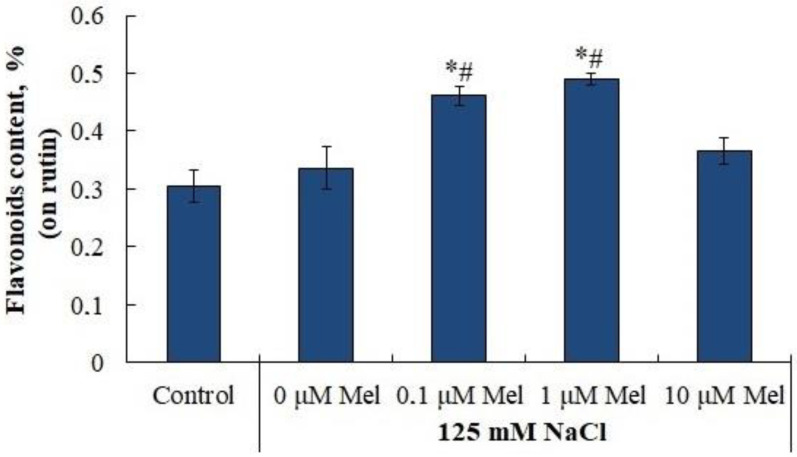
Influence of melatonin and chloride salinity on flavonoids content in *Solanum tuberosum* leaves. Asterisks (*) indicate significant differences from the control variant and the «#» indicates significant differences from the «125 mM NaCl» variant (*p* < 0.05).

**Table 1 ijms-24-06134-t001:** Influence of melatonin and chloride salinity on the growth rates of *Solanum tuberosum* plants.

Treatment		Stem Length, cm	Root Length, cm	Total Leaf Area, cm^2^	Number of Stolons, Unit	Total Weight of the Plant, g
NaCl, mM	Melatonin, µM					
0	0	10.00 ± 0.46	16.60 ± 0.73	15.55 ± 1.59	2.33 ± 0.37	2.47 ± 0.18
125	0	8.00 ± 0.26 *	13.47 ± 0.13 *	5.34 ± 0.34 *	0.39 ± 0.13 *	1.49 ± 0.10 *
125	0.1	9.17 ± 0.31	12.43 ± 0.72 *	4.22 ± 0.43 *	1.43 ± 0.37 *^,#^	1.50 ± 0.14 *
125	1.0	7.89 ± 0.20 *	14.30 ± 0.96 *	4.70 ± 0.45 *	1.50 ± 0.34 *^,#^	1.49 ± 0.10 *
125	10.0	8.63 ± 0.50	13.00 ± 0.65 *	4.83 ± 0.28 *	1.50 ± 0.33 *^,#^	1.59 ± 0.08 *

* *p* < 0.05 compared to the control value; # *p* < 0.05 compared to «125 mM NaCl» value.

**Table 2 ijms-24-06134-t002:** Effect of melatonin and chloride salinity on the accumulation of inorganic ions in different parts of *Solanum tuberosum* plants.

Treatment		Ion Content in Different Parts of the Plant					
NaCl, mM	Melatonin, µM	Leaves	%	Stems	%	Roots	%
		**Na^+^, mg/g (DW)**					
0	0	1.10 ± 0.10	100	2.55 ± 0.21	100	1.51 ± 0.06	100
125	0	81.44 ± 0.25 *	7404	81.94 ± 3.58 *	3213	39.51 ± 1.48 *	2617
125	0.1	52.01 ± 0.58 *^,#^	4728	45.35 ± 2,81 *^,#^	1778	44.64 ± 4.91 *	2956
125	1.0	25.53 ± 3.36 *^,#^	2321	56.52 ± 3.53 *^,#^	2216	28.87 ± 1.12 *^,#^	1912
125	10.0	35.54 ± 1.54 *^,#^	3231	58.73 ± 2.12 *^,#^	2303	27.68 ± 2.88 *^,#^	1833
		**Cl^−^, mg/g (DW)**					
0	0	8.60 ± 1.01	100	7.91 ± 0.46	100	10.26 ± 0.85	100
125	0	59.50 ± 6.89 *	692	108.49 ± 7.64 *	1372	55.69 ± 3.29 *	543
125	0.1	59.67 ± 5.17 *	694	70.94 ± 2.35 *^,#^	897	75.79 ± 6.14 *^,#^	739
125	1.0	44.80 ± 2.80 *^,#^	521	69.70 ± 4.72 *^,#^	881	36.72 ± 3.51 *^,#^	358
125	10.0	61.47 ± 6.19 *	715	82.40 ± 2.75 *^,#^	1042	40.83 ± 3.62 *^,#^	398
		**K^+^, mg/g (DW)**					
0	0	50.33 ± 1.33	100	55.08 ± 2.06	100	52.50 ± 2.90	100
125	0	29.16 ± 0.34 *	58	39.30 ± 1.99 *	71	38.90 ± 3.81 *	74
125	0.1	29.65 ± 0.97 *	59	50.48 ± 3.50 ^#^	92	53.60 ± 5.40 ^#^	102
125	1.0	34.39 ± 3.33 *^,#^	68	45.50 ± 2.35 *	83	54.60 ± 2.60 ^#^	104
125	10.0	46.31 ± 2.03 *^,#^	92	57.78 ± 2.63 ^#^	105	32.50 ± 2.31 *	62

Asterisks (*) indicate significant differences from the control variant and the «#» indicates significant differences from the «125 mM NaCl» variant (*p* < 0.05).

**Table 3 ijms-24-06134-t003:** Effect of melatonin on selective interorgan transport of ions by *Solanum tuberosum* plants under saline conditions.

Treatment		S_K^+^_ _Na^+^_ (K^+^/Na^+^ in the Leaves)/ (K^+^/Na^+^ in the Roots)	S_K^+^_ _Na^+^_ (K^+^/Na^+^ in the Stems)/ (K^+^/Na^+^ in the Roots)
NaCl, mM	Melatonin, µM		
0	0	8.98 ± 0.65	4.27 ± 0.28
125	0	0.35 ± 0.028 *	0.49 ± 0.03 *
125	0.1	0.45 ± 0.031 *	0.87 ± 0.05 *^,#^
125	1.0	0.71 ± 0.055 *^,#^	1.04 ± 0.09 *^,#^
125	10.0	1.11 ± 0.09 *^,#^	0.84 ± 0.06 *^,#^

Asterisks (*) indicate significant differences from the control variant and the «#» indicates significant differences from the «125 mM NaCl» variant (*p* < 0.05).

**Table 4 ijms-24-06134-t004:** Influence of melatonin on the levels of transcripts of genes encoding Na^+^/H^+^ antiporters of the tonoplast (*NHX1* to *NHX3*) and plasmalemma (*SOS1*) in *Solanum tuberosum* plants under saline conditions.

Treatment		*NHX1*, Rel. Units	%	*NHX2*, Rel. Units	%	*NHX3*, Rel. Units	%	*SOS1*, Rel. Units	%
NaCl, mM	Melatonin, µM								
0	0	0.0049 ± 0.0004	100	0.0006 ± 0.0001	100	0.0008 ± 0.0001	100	0.00016 ± 0.000017	100
125	0	0.0054 ± 0.0003	110	0.0014 ± 0.0001 *	233	0.0026 ± 0.0001 *	325	0.00027 ± 0.000024 *	164
125	0.1	0.0050 ± 0.0002	101	0.0015 ± 0.0003 *	250	0.0026 ± 0.0003 *	325	0.00021 ± 0.000013 *	127
125	1.0	0.0036 ± 0.0002 *^,#^	74	0.0010 ± 0.0001 *	155	0.0022 ± 0.0003 *	275	0.00026 ± 0.000015 *	158
125	10.0	0.0037 ± 0.0004 *^,#^	76	0.0012 ± 0.0002 *	195	0.0026 ± 0.0002 *	325	0.00034 ± 0.000032 *	211

Asterisks (*) indicate significant differences from the control variant and the «#» indicates significant differences from the «125 mM NaCl» variant (*p* < 0.05).

## Data Availability

The data supporting the findings of this study are available within the article in its Appendix A.

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
