# Peer review of "Priming Potato Plants with Melatonin Protects Stolon Formation under Delayed Salt Stress by Maintaining the Photochemical Function of Photosystem II, Ionic Homeostasis and Activating the Antioxidant System"

_ijms, 2023, doi:10.3390/ijms24076134_

Round 1

Reviewer 1 Report

In the paper authors studied the effect of melatonin priming on potato plants under salt stress at the molecular level and at the level of whole plants. Methods and results are well presented and discussed. At the same time, no clear conclusion was made about the relationship between macro- and microparameters’ changes. It is important to give authors’ opinion on this issue.

Majority of molecular level parameters was protected to varying degrees from the effects of salt stress by melatonin priming, while the growth parameters mostly did not respond to melatonin treatment. What could be a reason of this phenomenon? Could it be the case that the growth parameters will respond at later stage of plant development?

Other remarks

L 244-247 There is a statement that pretreatment of plants with melatonin did not prevent the dramatic suppression of CHS1α gene. An explanation is desirable why in this situation an increase of flavonoids takes place.

L 334-336 The explanation of inability of melatonin priming to reduce negative effect on growth parameters by species-specificity is not convincing, it could be a result of inadequate conditions of comparison with [50-53].

L 81 “are [25].” - “are” should be omitted.

L 127 age of plants should be indicated.

L 400,401 “[60 - Ma  et al 2018].” – “- Ma  et al 2018”  should be omitted.

L 510, 511 “…upon  heating Buege and Aust (1978) [66].” – “Buege and Aust (1978)” should be omitted.

L 528 “Determine the optical density of solution B after….” better replace with "The optical density of solution B was determined after…”

In the table S4 the value of transcript level of CHS1α in plants primed with melatonin, µM 0,1 is presented as “0.0119 ± 0.0002” and “17%”in percent. One of the numbers is wrong.

 There are two types of presentation of the same item in the text: “Photosystem II” and “Photosystem 2”. Better use one.

Although the IJMS Instruction for authors consider the section Conclusions in research manuscript as not mandatory, in the present paper it is necessary to conclude the discussion with the generalized view upon the obtained results. 

Author Response

Responses to reviewers' comments

Dear Colleagues,

We read the comments by the reviewers, carefully. All three reviewers recognized that our manuscript is interesting but each made some comments about ways in which it could be improved. We have revised the manuscript, incorporating almost all of these comments.

In addition we are very thankful to all three reviewers for their comments, kind help and advice.

First of all, the authors sincerely thank reviews for their constructive and benevolent critical comments aimed at correcting the deficiencies found in the text. The criticism expressed by the reviewers was extremely helpful, as it significantly improved the manuscript.

Reviewer 1

  1. Majority of molecular level parameters was protected to varying degrees from the effects of salt stress by melatonin priming, while the growth parameters mostly did not respond to melatonin treatment. What could be a reason of this phenomenon? Could it be the case that the growth parameters will respond at later stage of plant development?

Answer: We are grateful to the reviewer for this interesting question. The assumption made by the reviewer may well be realised.  Hormones and melatonin are specialized metabolic regulators. Their biological effect depends on many factors, such as plant ontogenetic state, organ and tissue specificity, hormonal status, intensity and duration of a stressor, in particular salt stress, etc. Changes in any of these factors, e.g. stage of ontogenesis or plant age, may lead to the realization of the protective effect of melatonin not only at the level of stolon formation, but also at the level of other growth processes. It cannot be excluded that the protective effect of melatonin under saline conditions is realized primarily at the level of stolon formation, which is of key importance, in contrast to biomass accumulation or increase of axial organ size, for the subsequent vegetative reproduction of potatoes.

                               The following sentence has been added to the discussion (Section 4.1.): “The experimental data we obtained do not allow us to exclude that exogenous melatonin will show a salt-protective effect on potato plants at a different stage of ontogenesis or under different experimental conditions”.

  1. L 244-247 There is a statement that pretreatment of plants with melatonin did not prevent the dramatic suppression of CHS1α. An explanation is desirable why in this situation an increase of flavonoids takes place.

Answer: In prokaryotes, there is a strong correlation between the level of mRNA and the amount of protein encoded by this mRNA In eukaryotes, such a correlation is the exception rather than the rule. It is related to very complex regulation of eukaryotic gene expression, which is realized at the levels of splicing and processing of transcripts, translation and posttranslational modifications of proteins encoded by them. The presence of high or low levels of transcripts of a particular gene, such as the CHS1α gene, only suggests that flavonoids may be synthesised in the plant in high or low levels. In our case, there is no stimulation of flavonoid biosynthesis gene expression, but perhaps melatonin strongly inhibits the degradation of flavanoids. Decreased rate of flavonoid degradation may be accompanied by flavonoid accumulation, despite suppressed expression of the key flavonoid biosynthesis gene.

                        The following sentence has been added to the result section (Section 2.6.): The increase in flavonoid levels in potato plants under salt stress with inhibition of CHS1α expression can be explained by melatonin inhibition of flavonoid degradation under these conditions.

  1. L 334-336 The explanation of inability of melatonin priming to reduce negative effect on growth parameters by species-specificity is not convincing, it could be a result of inadequate conditions of comparison with [50-53].

Answer: We cannot but agree with the reviewer that the protective effect of melatonin on growth processes under stress has been demonstrated in different plant species. This does not exclude the species-specific protective effect of melatonin, but also does not fully explain our findings. We have a long experience with melatonin. The main feature of this regulator, in contrast to classical phytohormones, is the high variability of the experimental data. This indicates a significant lability of plant response to melatonin, which is determined by many, including unaccounted for, factors of different nature. These factors can be both the physiological state of the plant and the concentrations of melatonin active, as well as the intensity of the repressive action.

                          The following sentence has been added to the Discussion (Section 3.1.):  The inability of melatonin priming to reduce the subsequent negative effects of salt stress on potato plant growth is obviously species-specific, and may depend on the physiological status of the plant, the intensity and duration of the salt stress and the method of hormonal treatment, as pretreatment of oat plants to salt stress…”.

  1. L 127 age of plants should be

Answer: Thank you for your comment. Section 4.1 indicates that the plants were 51 days old.

  1. L 400,401 “[60 - Ma  et al 2018].” – “- Ma  et al 2018”  should be omitted.

Answer: We are grateful for correction of typos and simple mistakes. It is corrected.

  1. L 510, 511 “…upon  heating Buege and Aust (1978) [66].” – “Buege and Aust (1978)” should be omitted.

Answer: It is corrected.

  1. L 528 “Determine the optical density of solution B after….” better replace with "The optical density of solution B was determined after…” .

Answer: It is corrected.

  1. In the table S4 the value of transcript level of CHS1α in plants primed with melatonin, µM 0,1 is presented as “0,0119±0,0002” and “17%”in percent. One of the numbers is wrong.

Answer: It is corrected.

  1. There are two types of presentation of the same item in the text: “Photosystem II” and “Photosystem 2”. Better use one.

Answer: Thank you, this typo was corrected.

  1. Although the IJMS Instruction for authors consider the section “Conclusions” in research manuscript as not mandatory, in the present paper it is necessary to conclude the discussion with the generalized view upon the obtained results. 

Answer: The authors gratefully accepted this suggestion from the reviewer. The conclusion is written and placed in the relevant section of the manuscript.

  1. Conclusions

When applied to salt-stressed potato plants, exogenous melatonin exerted clear stress-protective effects on the molecular level, maintaining plant water status and photosynthetic activity and increasing the accumulation of stress-protective compounds. However, these protective effects did not result in the maintenance of growth parameters of melatonin-treated plants compared to non-treated ones. The only clear exception was the formation of stolons, which was enhanced greatly by melatonin treatment. We can speculate that the protective effects of melatonin on the molecular level aided to maintain the carbohydrate supply, which maintained the stolon formation and hence the potential tuber productivity. More in-depth studies of potato carbon balance under salt stress and melatonin treatment are required to test this hypothesis.    

Reviewer 2 Report

The MS is very interesting and extensive. 

I suggest to introduce some recent refs. to discuss in flavonoid and osmotic sections, such as: doi:10.3390/ijms232315217 and https://doi.org/10.1093/jxb/erac009;

A question to resolve refers to the MEL concn. range selection: why not higher ones?

Author Response

Dear Colleagues,

We read the comments by the reviewers, carefully. All three reviewers recognized that our manuscript is interesting but each made some comments about ways in which it could be improved. We have revised the manuscript, incorporating almost all of these comments.

In addition we are very thankful to all three reviewers for their comments, kind help and advice.

First of all, the authors sincerely thank reviews for their constructive and benevolent critical comments aimed at correcting the deficiencies found in the text. The criticism expressed by the reviewers was extremely helpful, as it significantly improved the manuscript.

Reviewer 2

  1. I suggest to introduce some recent refs. to discuss in flavonoid and osmotic sections, such as: doi:3390/ijms232315217 and https://doi.org/10.1093/jxb/erac009;

Answer: Thank you for this offer.Two suggested references are now included.

  1. A question to resolve refers to the MEL concn. range selection: why not higher ones?

Answer: Thank you for this question. I can inform you that the selection of effective concentrations of melatonin was carried out in preliminary experiments. The results showed that a further increase in the used melatonin concentration by several times did not lead to any significant changes of the tested physiological parameters. Furthermore, in our experiments, the primary site of action of melatonin is the root system, as we introduced melatonin into the nutrient medium. It is known that the root, unlike the above-ground part of plants, is highly sensitive to the action of exogenous regulators. As is known from literature data, high concentrations of melatonin can exhibit not only a stimulating, but also an inhibitory effect. The use of melatonin in low concentrations for practical purposes is preferable from an ecological and economic point of view.

Reviewer 3 Report

There are comments to the article text in question. 

2. Results

In the numerical designations on the axes of figures, "comma" should be replaced by "point ".

Р. 12. Line 291.

Table 3: The fourth header has no division between the data in the two brackets. 

(K+ /Na+ in the stems) (K+ /Na+ in the roots) replace with (K+ /Na+ in the stems) / (K+ /Na+ in the roots)

Р. 12. Line 307-309.

There is a statement in the text about the increase in SOS1 gene transcripts, but there is no sign of significant parameter (“125 μM NaCl” and “125 μM NaCl+10 μM Mel”) differences in Table 4.

3. Discussion. 

Р. 14. Line 319-322. Insert a link to authors studying the problem.

Р. 15. Line 380-383. This is a verbatim quote with a reference to the original source, but without a transcription of the designations (МТ, TAG, FA и PM).

Р. 15. Line 400-401. [60 - Ma 400 et al 2018]. Remove the author from the link.

Article Title - Discussion. There is no complete correspondence between the title of the article and the Discussion on the regulation by melatonin of stolon formation.

The authors highlight stolons in the article title, so they should discuss the role of melatonin in their formation in saline conditions in the Discussion. Otherwise, the title of the article should be changed.

4. Methods

Р. 17. Line 488-489. In the Methods for the determination of photosynthetic pigments (4.5) it is not necessary to specify decimal wavelengths, because in the specifications of the Spectrophotometer Genesys 10S UV-Vis, Thermo: reproducibility of wavelength setting, nm - ±0.5; error in wavelength setting, nm - ±1.

Р. 18. Line 518. 4.9. Determination of the total content of flavonoids. There is no reference to the author of the method.

Р. 19. Line 567. Correct the typo in the formula:

 (K+ /Na+ in the shoots or leaves) (K+ /Na+ in the roots) replace with (K+ /Na+ in the shoots or leaves) / (K+ /Na+ in the roots).

Р. 19. Line 568. Remove formula numbering: (1).

Р. 19. 4.14. The selection of target genes for qRT-PCR analysis and primer design

Line 577. There is no reference number for References.

Author Response

Dear Colleagues,

We read the comments by the reviewers, carefully. All three reviewers recognized that our manuscript is interesting but each made some comments about ways in which it could be improved. We have revised the manuscript, incorporating almost all of these comments.

In addition we are very thankful to all three reviewers for their comments, kind help and advice.

First of all, the authors sincerely thank reviews for their constructive and benevolent critical comments aimed at correcting the deficiencies found in the text. The criticism expressed by the reviewers was extremely helpful, as it significantly improved the manuscript.

Reviewer 3

  1. In the numerical designations on the axes of figures, "comma" should be replaced by "point ".

Answer: We thank the reviewer for his comment. All drawings have been carefully reviewed and the technical corrections indicated have been made.

  1. Р. 12. Line 291.

Answer: It is corrected.

  1. Table 3: The fourth header has no division between the data in the two brackets. (K+ /Na+ in the stems) (K+ /Na+ in the roots) replace with (K+ /Na+ in the stems) / (K+ /Na+ in the roots).

Answer: Thank you. It is corrected.

  1. Р. 12. Line 307-309. There is a statement in the text about the increase in SOS1 gene transcripts, but there is no sign of significant parameter (“125 μM NaCl” and “125 μM NaCl+10 μM Mel”) differences in Table.

Answer: Thank you for your fair comment. It is corrected.

  1. Р. 14. Line 319-322. Insert a link to authors studying the problem.

Answer: Thank you for your useful offer. We used the article Kumar P., Sharma P. K. Soil salinity and food security in India. Frontiers in Sustainable Food Systems VOL 4 2020 URL=https://www.frontiersin.org/articles/10.3389/fsufs.2020.533781 DOI=10.3389/fsufs.2020.533781.

  1. Р. 15. Line 380-383. This is a verbatim quote with a reference to the original source, but without a transcription of the designations (МТ, TAG, FA и PM).

Answer: Thank you. It is corrected.

  1. Р. 15. Line 400-401. [60 - Ma 400 et al 2018]. Remove the author from the link.

Answer: Thank you for pointing out a technical error. It is corrected.

  1. Article Title - Discussion. There is no complete correspondence between the title of the article and the Discussion on the regulation by melatonin of stolon formation. The authors highlight stolons in the article title, so they should discuss the role of melatonin in their formation in saline conditions in the Discussion. Otherwise, the title of the article should be changed.

Answer: We thank the reviewer for the point made, but we cannot agree that the discussion does not correspond to the title of the article or that the title of the article does not correspond to the discussion presented. First of all, in section 3.1. of the discussion, we discuss the reasons for the negative effect of salt stress on stolon formation, using the literature data, and the reasons for the possible protective effect of melatonin on this process under salt stress. In addition, we found that exogenous melatonin shows a protective effect only on the process of stolon formation at the whole plant level. The entire subsequent experimental part of the work, as well as the entire discussion, was aimed at elucidating the physiological mechanisms that are regulated by melatonin and possibly underlie its protective effect on stolon formation.

  1. Р. 17. Line 488-489. In the Methods for the determination of photosynthetic pigments (4.5) it is not necessary to specify decimal wavelengths, because in the specifications of the Spectrophotometer Genesys 10S UV-Vis, Thermo: reproducibility of wavelength setting, nm - ±0.5; error in wavelength setting, nm - ±1.

Answer: Thank you for pointing out a technical error. It is corrected.

  1. Р. 18. Line 518. 4.9. Determination of the total content of flavonoids. There is no reference to the author of the method.

Answer: Thank you. The necessary reference to the method of analysis of flavonoids used is included in the text and reference list (Gage, T.B.; Wendei, S.H. Quantitative determination of certain flavonol 3-glycosides. Anal. Chem. 1950, 22, 708-711).

  1. Р. 19. Line 567. Correct the typo in the formula: (K+ /Na+ in the shoots or leaves) (K+ /Na+ in the roots) replace with (K+ /Na+ in the shoots or leaves) / (K+ /Na+ in the roots).

Answer: It is corrected.

  1. Р. 19. Line 568. Remove formula numbering: (1).

Answer: It is corrected.

  1. Р. 19. 4.14. The selection of target genes for qRT-PCR analysis and primer design

Answer: К сожалению, нам не понятно, что имел в виду рецензент, давая ссылку на подзаголовок 4.14. раздела методов.

  1. Line 577. There is no reference number for References. There is reference number for References from Nicot et al. (2005). Reference 72. Previously mentioned in paragraph 4.15., now in 4.14.

Answer: It is corrected.
